# Influence of Concentration, Surface Charge, and Natural Water Components on the Transport and Adsorption of Polystyrene Nanoplastics in Sand Columns

**DOI:** 10.3390/nano14060529

**Published:** 2024-03-15

**Authors:** Gabriela Hul, Hande Okutan, Philippe Le Coustumer, Stéphan Ramseier Gentile, Stéphane Zimmermann, Pascal Ramaciotti, Pauline Perdaems, Serge Stoll

**Affiliations:** 1Department F.-A. Forel for Environmental and Aquatic Sciences, Institute for Environmental Sciences, University of Geneva, 1205 Geneva, Switzerland; 2Ecole Doctorale, Université de Bordeaux Montaigne, 33607 Pessac, France; 3Department of Geological Engineering, University of Mugla Sitki Kocman, Mugla 48260, Türkiye; 4Bordeaux Imaging Center CBRS—INRAE—INSERM, Université de Bordeaux, 33000 Bordeaux, France; 5SIG—Industrial Services of Geneva, 1211 Geneva, Switzerland

**Keywords:** nanoplastics, porous media, mobility, retention, aggregation, straining, NOM

## Abstract

Information about the influence of surface charges on nanoplastics (NPLs) transport in porous media, the influence of NPL concentrations on porous media retention capacities, and changes in porous media adsorption capacities in the presence of natural water components are still scarce. In this study, laboratory column experiments are conducted to investigate the transport behavior of positively charged amidine polystyrene (PS) latex NPLs and negatively charged sulfate PS latex NPLs in quartz sand columns saturated with ultrapure water and Geneva Lake water, respectively. Results obtained for ultrapure water show that amidine PS latex NPLs have more affinity for negatively charged sand surfaces than sulfate PS latex NPLs because of the presence of attractive electrical forces. As for the Geneva Lake water, under natural conditions, both NPL types and sand are negatively charged. Therefore, the presence of repulsion forces reduces NPL’s affinity for sand surfaces. The calculated adsorption capacities of sand grains for the removal of both types of NPLs from both types of water are oscillating around 0.008 and 0.004 mg g^−1^ for NPL concentrations of 100 and 500 mg L^−1^, respectively. SEM micrography shows individual NPLs or aggregates attached to the sand and confirms the limited role of the adsorption process in NPL retention. The important NPL retention, especially in the case of negatively charged NPLs, in Geneva Lake water-saturated columns is related to heteroaggregate formation and their further straining inside narrow pores. The presence of DOM and metal cations is then crucial to trigger the aggregation process and NPL retention.

## 1. Introduction

Nanoplastics (NPLs), described as plastic particles comprised between 1 nm and 1 µm, are a new type of nanomaterial [1,2]. Depending on the origin, NPLs are divided into categories of either primary or secondary NPLs. Primary NPLs are intentionally produced in the nanosize range and used in cosmetic products or in abrasive blasting [3,4]. As for the secondary NPLs, they are derived from biodegradation, photodegradation, thermo-oxidative degradation, thermal degradation, or hydrolysis of larger plastic debris [5]. The production of plastics has been progressively growing since the 1950s and will reach 367 million tons in 2020 [6]. The increased use of plastics in everyday life and inadequate waste management practices led to the accumulation of plastic debris in the natural environment, especially in aquatic systems. Plastic particles have been detected worldwide, in marine systems [7], rivers [8,9], and even in remote alpine lakes [10]. Nanoplastics (NPLs) pollution in freshwater systems is estimated to be more serious than in marine environments [11,12,13]. Nanoplastics pose a serious threat for human beings, as they can diffuse inside the body, accumulate in different organs and tissues, and induce inflammatory reactions or cancer development [14]. Since freshwater resources are often used to produce drinking water, it is crucial to understand NPL behavior during conventional treatment, especially during filtration processes, to protect consumers health [15,16,17,18]. Regarding laboratory scale models, saturated columns are usually used to mimic NPLs and MPLs [19] behavior in soils, sediments, aquifers, and industrial filtration units. For instance, Pradel et al. (2020) investigated the transport of environmentally relevant negatively charged carboxylated polystyrene latex NPLs in sand columns and found that NPL shape plays a key role in NPL retention inside porous media [20]. Interestingly, Ma et al. (2020) did not observe a difference in retention between aged rod-shaped and spherical nanoplastics and pointed out that NPL retention was strongly influenced by colloidal surface properties and hydrodynamics [21]. Among the physicochemical and hydrodynamic factors controlling NPL transport, flowrate, salinity, temperature, pH, ionic strength, and porous media composition seem to play an important role. Ling et al. studied the vertical transport and retention behavior of negatively charged polystyrene NPLs in a simulated hyporheic zone and reported that high flowrate, low salinity, high water saturation, and low temperature facilitated NPL mobility [22]. Moreover, Zhang et al. (2023) investigated the transport of negatively charged polystyrene NPLs in saturated quartz sand columns and reported increased NPL mobility with an increase in input concentration, flowrate, and sand grain size. In addition, Zhang et al. (2023) studied the transport of both positively and negatively charged polystyrene NPLs in binary metal oxide-saturated porous media and reported that the presence of Fe/Al oxides and high ionic strengths prevent NPLs from flowing out of porous media [23]. 

Regarding real-scale experiments, pilot-scale filters are used to simulate NPL behavior during conventional drinking water treatment. For example, Ramirez et al. (2022) evaluated the removal efficiency of positively charged polystyrene nanoplastics during the sand and activated carbon filtration processes and reported that overall NPL removal was approximately 88% and retention was mostly controlled by adsorption and straining [24]. Pulido-Reyes et al. (2022) also investigated nanoplastics removal during drinking water treatment and compared laboratory and pilot-scale experiments [25]. They found that NPL retention inside sand laboratory columns saturated with ultrapure water varied with flow conditions and column length. For example, for the longest column (20 cm), NPL retention was equal to 60% and 40% for flowrates of 5 mL min^−1^ and 21 mL min^−1^, respectively. Moreover, they reported that the C/C_0_ value, calculated for flowrates of 5 mL min^−1^ and a medium column (11.5 cm in length) filled with pristine sand, was equal to 57%. They observed that further changes in filtration media from pristine sand to aged sand decreased the C/C_0_ value by up to 23% for the same operational conditions. They also noticed that the further change of feed water from ultrapure water to Lake Zurich water contributed to the almost complete removal of NPLs. Pilot-scale experiments confirmed their laboratory findings and showed that NPLs were mostly concentrated at the top of the sand filter, up to 0.9 m in depth. 

Although these studies provided valuable information on NPLs retention and transport in porous media, the influence of NPLs surface charge on their transport and the influence of high NPL concentrations (>100 mg L^−1^) on the retention capacities of porous media have received less attention. In addition, batch experiments revealed an important role of the adsorption process in nanoplastics and engineered nanoparticle removal; however, changes in the adsorption capacities of porous media under dynamic conditions and in the presence of other retention mechanisms are still not well understood [26]. In this context, the main objective of this study concerns the transport of highly concentrated nanoplastics with different functional groups in porous media saturated with different types of water. Polystyrene latex NPLs with positive amidine and negative sulfate functional groups were selected as model NPLs because of their extensive applications, use in many laboratory experiments, and frequent occurrence in the natural environment [27,28]. Untreated quartz sand, used by the main drinking water treatment plant in Geneva (Switzerland), was considered a proxy for porous media. Sand grains contained trace quantities of clay minerals and metal oxides and did not receive any special treatment to better represent subsurface environments and conditions prevailing during drinking water production. The laboratory column experiments were conducted to examine the transport behavior of high concentrations (100 mg L^−1^ and 500 mg L^−1^) of both NPL types in quartz sand columns saturated with two types of water: ultrapure water and Geneva Lake water. Quartz sand columns saturated with ultrapure water were considered a reference case to better explain the role of natural water components in NPL transport through porous media. Dynamic light scattering, laser Doppler velocimetry, and SEM imaging were used to characterize NPLs in effluent and different types of water and to gather insights into the mechanisms governing the transport and retention of NPLs in porous media.

## 2. Materials and Methods

### 2.1. Materials

#### 2.1.1. Polystyrene Latex Nanoplastics

Surfactant-free polystyrene latex nanoplastics (NPLs) with positive amidine functional groups (–NH–NH_2_^+^) on their surfaces were synthesized by the Institute of the Analytical Sciences (ISA, Lyon, France). As communicated by the provider, the styrene concentration in the provided suspension was equal to 50 g L^−1^, and the conversion into polystyrene was estimated to be 86%. The z-average hydrodynamic diameter of amidine PS latex NPLs was verified by dynamic light scattering, and it was equal to 189 ± 1.5 nm. Uncoated sulfate PS latex NPLs with negative sulfate functional groups $\left(-{SO}_{4}^{2-}\right)$ on their surfaces were purchased from Invitrogen (ThermoFisher Scientific, Reinach, Switzerland). According to the manufacturer, they are characterized by a primary diameter of 0.1 µm, a density equal to 1.055 g cm^−3^ at 20 °C, and a specific surface area of 5.9 × 10^5^ cm^2^ g^−1^. The size of sulfate PS latex NPLs was also verified by dynamic light scattering, and it was equal to 115 ± 1.5 nm. Both stock solutions were kept in the dark at low temperatures and were subsequently used to prepare dilutions. 

pH-titration curves were established for both types of NPLs, and a detailed description of the procedure can be found in Appendix A. The stability of NPLs in Geneva Lake water was also evaluated. Moreover, 100 mg L^−1^ and 500 mg L^−1^ NLP suspensions were prepared directly in Geneva Lake water, and the evolution of NPLs z-average hydrodynamic diameter and ζ potential was followed up to 120 min with ZetaSizer Nano ZS, Malvern, UK. 

#### 2.1.2. Ultrapure and Geneva Lake Water

The following two types of water were used to prepare NPL solutions and during filtration experiments: ultrapure water and Geneva Lake water. Ultrapure water was produced in the laboratory using the Milli-Q water system (Milli-Q water, Millipore, Switzerland, with a R > 18 MΩ·cm and a total organic carbon (TOC) < 2 ppb). Geneva Lake water was chosen as an environmental matrix because it is used to produce drinking water for over 500,000 consumers in the canton of Geneva (Switzerland). It was sampled in Geneva, near the Rhone River estuary (46°12′06.1″ N 6°07′23.7″ E). Its ionic composition was determined using ion chromatography (ICS 3000, Dionex, Reinach, Switzerland), its total organic carbon (TOC) content was determined with a TOC-L analyzer (Shimadzu, Kyoto, Japan), and its pH and conductivity values were measured using the Hach Lange HQ40d portable meter. The results are presented in Appendix A. The water samples were stored in a dark place at a temperature of 4 °C for no longer than four weeks. Prior to analyses, they were filtered using a cellulose membrane filter with a 0.45 µm pore size to remove impurities that could affect turbidity measurements. 

#### 2.1.3. Quartz Sand

Quartz sand samples were provided by Carlo Bernasconi AG (Zürich, Switzerland), and they contain from 97% to 99% silica and trace amounts of Al_2_O_3_, K_2_O, Na_2_, TiO_2_, Fe_2_O, TiO, CaO, MgO, Fe_2_O_3_, and Na_2_O. The size and morphology characteristics of sand grains determined using Camsizer (Retsch, Haan, Germany) were measured by the Industrial Services of Geneva and are presented in Appendix A. SEM images of the pristine sand surface are presented in Appendix A.

### 2.2. Methods

#### 2.2.1. Laboratory Column Experiments

Column experiments were performed in duplicate at room temperature using acrylic columns (36 mm inner diameter, 200 mm length) sealed with steel screens and caoutchouc stoppers to prevent any sand loss. The columns were wet-packed with quartz sand grains and gently tapped during the filling procedure to avoid any entrapped air and to help the stationary phase spread evenly. An upward flow of ultrapure or Geneva Lake water was induced within the column using a peristaltic pump (Minipuls 3, Gilson, Emmen, Switzerland). A flowrate was set to 20.0 ± 2.0 mL min^−1^ for each experiment to mimic rapid sand filtration [25]. In the drinking water treatment plant of Geneva (Switzerland), rapid sand filtration is also used, and the average filtration speed is 6 m h^−1^. The pH of ultrapure water was equal to 5.7 ± 0.2, whereas the pH of Geneva Lake water was equal to 7.2 ± 0.2. As the pH in both cases was stable during the experiments, there were no additional adjustments. An experimental setup is presented in Appendix A. First, the prepared columns were conditioned by introducing ultrapure or Geneva Lake water at 20.0 ± 2.0 mL min^−1^ for 31–34 pore volumes (PV). One pore volume is the cross-sectional area of the column times the length times the porosity, i.e., the column’s capacity [29]. In this study, 1 PV ranged between 69 mL and 73.0 mL. The number of pore volumes varied slightly as a function of applied flowrate and sand height. Subsequently, either fluorescent tracer solution or NPL solution were injected into the column, followed by the background solution. The type of injection can be defined as a short-pulse or instantaneous injection. The samples were collected every 30 s in the case of fluorescent tracer, every 60 s in the case of NPL solutions, and every 30 s after the injection. After each NPL transport experiment, a 1 cm layer of sand was collected at the bottom of the column for further analysis. Moreover, after each experiment, the pH of the effluent was controlled, but no significant changes were observed. Breakthrough curves (BTCs) of fluorescent tracer and NPLs were obtained by plotting the effluent concentration normalized by the concentration in the feed suspension (C/C_0_) as a function of the filtered water expressed in pore volumes. BTCs were used to calculate the recovered mass and adsorption capacity. Detailed information about these calculations can be found in Appendix A.

#### 2.2.2. Tracer Experiments

Tracer experiments, using fluoresceine sodium salt, were conducted to determine the porosity and hydrodynamic characteristics of porous media. A detailed description of these experiments can be found in Appendix A.

#### 2.2.3. NPL Characterization and Concentration Determination

Z-average hydrodynamic diameter and ζ potential of amidine PS latex NPLs and sulfate PS latex NPLs were determined with Zetasizer Nano ZS using dynamic light scattering (DLS) and laser doppler velocimetry (LDV), respectively. As for the NPL concentrations in the effluent, they were determined using a Hach Turbidimeter model TU5200 (Hach Lange, Rheineck, Switzerland). Detailed information about the methods used are presented in Appendix A.

#### 2.2.4. SEM Imaging

A JSM-7001FA (JEOL Ltd., Tokyo, Japan) scanning electron microscope (department of Earth Sciences, University of Geneva, Switzerland) was used to obtain secondary electron images of NPLs dispersed in different types of water as well as sand grains before and after contact with NPL solution. A detailed description of the preparation of SEM samples can be found in our previous work [30]. 

## 3. Results and Discussion

### 3.1. Characterization of Nanoplastics Dispersed in Ultrapure and Geneva Lake Water

Changes in z-average hydrodynamic diameter and ζ potential of 50 mg L^−1^ amidine PS latex NPLs dispersed in ultrapure water as a function of pH are presented in Appendix A. At a pH of 5.7 (ultrapure water), both z-average hydrodynamic diameter and ζ potential are stable and equal to 190 ± 5 nm and +50 ± 5 mV, respectively. SEM images obtained for these conditions showed NPLs in the form of dimers, trimers, and small aggregates (Appendix A.

Variations of the z-average hydrodynamic diameter and ζ potential of sulfate PS latex NPLs dispersed in ultrapure water as a function of pH are presented in Appendix A. Z-average hydrodynamic diameter is stable independently of pH and equal to 116 ± 1 nm. As for the ζ potential values, they are negative in the whole range of pH and vary between −34 mV and −48 mV. SEM images obtained for ultrapure water at a pH of 5.7 showed NPLs in the form of individuals or dimers (Appendix A).

The stability of 100 and 500 mg L^−1^ amidine PS latex NPLs dispersed in Geneva Lake water was evaluated over time (Appendix A). For 100 mg L^−1^ dispersion, the z-average hydrodynamic diameter is varying around 256 ± 4 nm, and the ζ potential is oscillating around +20.0 ± 0.5 mV. As for the 500 mg L^−1^ dispersion, the z-average hydrodynamic diameter is slightly lower and equal to 177 ± 3 nm, and the ζ potential is slightly higher and equal to +30.0 ± 0.6 mV. The stability of 100 and 500 mg L^−1^ sulfate PS latex NPLs dispersed in Geneva Lake water was also evaluated over time (Appendix A). There is no important difference in size or charge between 100 and 500 mg L^−1^ dispersions. Z-average hydrodynamic diameter ranges between 124 ± 1 nm and 112 ± 1 nm, whereas ζ potential ranges between −23 ± 1 mV and −24.9 ± 0.8 mV. Interestingly, despite the presence of DOM, metal ions, and other natural components, both types of nanoplastics are very stable and well dispersed. 

### 3.2. Nanoplastics Transport in Saturated Quartz Sand Columns 

Tracer column experiments were performed in the first place to determine the porosity and hydrodynamic characteristics of quartz sand columns. They indicated that the filter bed is characterized by a porosity of 0.41 ± 0.01, a mean velocity of 5.15 ± 0.02 cm min^−1^, and a dispersion coefficient of 2.00 ± 0.02 cm^2^ min^−1^.

#### 3.2.1. NPLs Transport and Retention in Ultrapure Water-Saturated Columns

Filtration experiments using sand columns saturated with ultrapure water were performed first. BTCs of both amidine PS latex and sulfate PS latex NPLs are presented in Figure 1a. Independently of the NPL type or concentration, NPLs start to leave the filtration column at 0.4 PV, and their flux continues up to 1.2 PV. For 100 mg L^−1^ and 500 mg L^−1^ amidine PS latex NPLs, maximal C/C_0_ values are equal to 0.002 and 0.0017, respectively. As for the 100 mg L^−1^ and 500 mg L^−1^ sulfate PS latex NPLs, maximal C/C_0_ values are higher and equal to 0.007 and 0.009, respectively. BTCs obtained for amidine PS latex have a descending segment that is quite elongated, whereas BTCs obtained for sulfate PS latex NPLs are very symmetrical. BTC shapes suggest that there was no interruption in the transport of sulfate PS latex NPLs, whereas some delay appeared in the case of amidine PS latex NPLs. As previously reported, the observed retardation may be due to the occurrence of reversible adsorption or the increased transport time of larger aggregates [25,31]. The transport of amidine PS latex NPLs is decreasing with an increase in concentration, whereas the transport of sulfate PS latex is increasing with an increase in concentration. The decrease in amidine PS latex NPL mobility can be attributed to the adsorption of positively charged NPLs to the negatively charged sand surface, whereas the increase in sulfate PS latex NPL transport is probably due to surface-active site saturation and the presence of repulsion forces between similarly charged NPLs and the sand surface. SEM images revealed that a small amount of amidine PS latex NPLs is attached to the sand, especially inside concaves or within surface irregularities, and to the clay minerals or metal oxide surfaces (Figure 2b). As for the sulfate PS latex NPLs, SEM imaging showed that they prefer to attach to bare sand within surface irregularities; however, the number of attached NPLs is not as high as in the case of amidine PS latex (Figure 2c). Surface heterogeneities are preferred as attachment sites because they are characterized by higher charge density and provide more available surface-active sites than a plain surface [32]. Moreover, surface irregularities can also act as an obstacle and physically block NPLs from flowing out of porous media, especially in the case when repulsion forces exist between NPLs and the filtration bed. Regarding clay minerals and metal oxides, they might be chosen as attachment sites because they are characterized by variable charge components, and, depending on the physicochemical conditions, they can provide additional surface-active sites for NPL deposition [33,34,35]. The importance of DLVO-type interactions and surface roughness in NPL transport and retention in porous media have already been revealed in previous studies [20,36,37,38]. The average adsorption capacity of sand grains for the removal of both NPL types was very small and equal to 0.008 mg g^−1^ for NPL concentrations of 100 mg L^−1^. By contrast, the average adsorption capacity of sand for the removal of 50 mg L^−1^ of amidine PS latex NPLs dispersed in ultrapure water was equal to 0.10 mg g^−1^ [30]. The much smaller adsorption capacity obtained in this study is probably due to shortened residence time and increased drag forces, which hindered NPL deposition on the sand surface and increased their mobility within the porous media.

The obtained BTCs were subsequently used to calculate the mass of NPLs retained inside porous media, and the results are presented in Figure 2a. For amidine PS latex NPLs, retained mass increases with increasing NPL concentrations, from 75.5 ± 0.2% to 85 ± 2%. By contrast, for sulfate PS latex NPLs, retained mass decreases with increasing NPL concentrations, from 31 ± 6% to 18 ± 1%. As can be seen, NPL retention in sand quartz columns is quite important, and adsorption is not the only process that can explain the observed results. Indeed, another retention mechanism could be straining, i.e., blocking of larger aggregates by narrow pores [39]. 

The size and charge of NPLs present in the effluent were evaluated in terms of z-average hydrodynamic diameter and ζ potential, respectively (Figure 1b,c). The z-average hydrodynamic diameter of amidine PS latex NPLs increased for both concentrations from 190 ± 3 nm (injection solution) and varies between 200 and 400 nm in the peak zone. In parallel, the ζ potential of amidine PS latex NPLs decreased for both solutions from +46 ± 3 mV (injection solution) and varies between +3 and +17 mV in the peak zone. Regarding the characteristics of recovered sulfate PS latex NPLs, the z-average hydrodynamic diameter slightly increased for both concentrations from 115 ± 1.5 nm and varies between 120 and 150 nm in the peak zone. Interestingly, aggregates ranging from 200 up to 700 nm were observed after the peak. However, as NPL concentrations after the peak were very low, the precision of these measurements may not be very high, and one must take these values with precaution. As for the ζ potential measurements, ζ potential decreased from −34.8 ± 0.6 mV (injection solution) and varies between 0 and −15 mV in the peak zone. Moreover, the ζ potential of aggregates observed after the peak oscillates around 0 mV. SEM images of the effluent confirmed NPLs presence and showed dimers, trimers, and larger aggregates (Appendix A). The above results suggest that NPLs confined in limited space have more chance to enter into contact and form aggregates, which can be further retained in narrow pores or in pores narrowed by previously deposited NPLs.

#### 3.2.2. NPLs Transport and Retention in Geneva Lake Water Saturated Columns

Filtration experiments using sand columns saturated with Geneva Lake water were conducted to assess the influence of natural water components, such as metallic ions and dissolved organic matter (DOM), on the transport of NPLs through sand filtration media. BTCs of both amidine PS latex and sulfate PS latex NPLs are presented in Figure 3a. In general, NPLs start to flow out at 0.4 PV, and their flux continues up to 1.2 PV. In the case of highly concentrated amidine PS latex NPLs, NPL transport is faster and finishes within 1 PV. For 100 and 500 mg L^−1^ amidine PS latex NPLs, maximal C/C_0_ values are equal to 0.002 and 0.007, respectively. As for the 100 and 500 mg L^−1^ sulfate PS latex, maximal C/C_0_ values are equal to 0.0004 and 0.003, respectively. BTCs obtained for both concentrations of sulfate PS latex and 100 mg L^−1^ amidine PS latex are rather elongated. However, the BTC obtained for the highly concentrated amidine PS latex solution is very symmetrical in comparison to three other BTCs. The shapes of descending segments of BTCs suggest that there was no significant tailing in the case of transport of highly concentrated amidine PS latex NPLs, whereas some retardation appeared in other cases. It means that there was no reversible adsorption in the case of highly concentrated amidine PS latex, and the interactions with the filter bed, which could delay the transport process, were limited, probably because of repulsion and surface-active site saturation. In other cases, reversible adsorption, NPL-sand interactions, and elongated transport of aggregates could be responsible for BTC shapes.

The transport of amidine PS latex NPLs is increasing with an increase in concentration, whereas the transport of sulfate PS latex seems not to be affected by the concentration increase. A decreased retention for higher concentrations of amidine PS latex NPLs may be due to saturation of surface-active sites and competition for negatively charged adsorption areas between NPLs and other positively charged water components. SEM images showed that amidine PS latex NPLs are attached to the sand surface as individuals and large aggregates, whereas a very low number of single sulfate PS latex NPLs are deposited in the vicinity of surface irregularities (Figure 4b–d). The average adsorption capacity of sand grains for the removal of both NPL types was found to be very small and equal to 0.007 mg g^−1^ for NPL concentrations of 100 mg L^−1^. By contrast, the adsorption capacity of sand for the removal of 50 mg L^−1^ of amidine PS latex NPLs dispersed in Geneva Lake water was equal to 0.7 ± 0.2 mg g^−1^ [30]. As mentioned before, smaller adsorption capacity is probably due to short residence times and the presence of drag forces. Obtained BTCs were also used to calculate the mass of NPLs retained inside porous media, and the results are presented in Figure 4a. For amidine PS latex NPLs, retained mass decreased with increasing NPL concentrations, from 80 ± 1.0% to 52 ± 1.5%. As for the sulfate PS latex NPLs, retained mass is relatively stable for both concentrations and equal to 70 ± 9% and 66 ± 7% for 100 mg L^−1^ and 500 mg L^−1^ solutions, respectively.

NPL aggregation and further aggregate entrapment can be responsible for increased NPL retention. Changes in the z-average hydrodynamic diameter and ζ potential of NPLs present in the effluent are shown in Figure 3b,c. The z-average hydrodynamic diameter of amidine PS latex NPLs increased for both concentrations from approximately 155–265 nm (injection solutions) and varies between 400 and 750 nm in the peak zone. Simultaneously, the ζ potential of amidine PS latex NPLs decreased for both solutions from approximately +21–+30 mV (injection solutions) and varies between −3 and −8 mV in the peak zone. As for the sulfate PS latex NPLs, the z-average hydrodynamic diameter increased from 116 ± 0.4 nm (injection solution) and varies from 350 to 690 nm in the peak zone for the 100 mg L^−1^ NPL solution. The size of highly concentrated sulfate PS latex NPLs increased slightly from 115 ± 2 nm (injection solution) up to 200–400 nm within the peak zone and up to 540 nm after the peak zone. In parallel, the ζ potential of the 100 mg L^−1^ NPL solution decreased from −26.1 ± 0.53 mV (injection solution) and varies between −2 and −5 mV within the peak zone. ζ potential of 500 mg L^−1^ NPL solution also decreased from −22.2 ± 0.35 mV (injection solution) and ranges between −5 and −13.5 mV in the peak zone and oscillates around −2 mV after the peak. SEM images of effluents showed NPL aggregates covered with natural organic matter for both types of plastics (Appendix A.

The aggregation process observed in this case is not only due to the reduced distance between NPLs but also to the presence of DOM particles and metal cations. DOM coating and DOM screening effects reduce electrostatic repulsion between NPLs and trigger the aggregation process. As for the metal cations, they act as electrostatic bridges between similarly charged NPLs and facilitate the aggregation process. Consequently, formed aggregates are too large to be transported through narrow pores and remain trapped inside. Apart from that, increased ionic strength can also lead to compression of the electric double layer of both NPLs and filtration media and cause limited deposition onto the mineral surface. Previous studies have already demonstrated that natural organic matter and metal ions can largely control NPLs and nanoparticle retention and transport in porous media [20,23,40,41,42,43].

### 3.3. Summary of Transport and Retention Mechanisms

Based on the results of column experiments, size measurements, ζ potential measurements, and SEM images, the transport, retention mechanisms, and properties of amidine PS latex and sulfate PS latex NPLs are summarized in Figure 5 to highlight the role of solution chemistry and the presence of natural compounds on NPLs transport and retention.

In ultrapure water, both amidine PS latex NPLs and sulfate PS latex NPLs injection solutions are stable due to the presence of repulsive forces (A). Once injected into the quartz sand columns, NPLs start to aggregate due to a limited pore space or adsorb to the sand surface (C, D). Aggregates can be further entrapped in narrow pores (C). As for the adsorption process, adsorption mechanisms are different for both NPLs types. Adsorption of amidine PS latex is due to the electrically attractive interactions between oppositely charged sand grains and NPLs. In addition, amidine PS latex NPLs adsorb in the vicinity of surface irregularities, probably because these zones are characterized by higher surface charge density. Despite the repulsion between similarly charged sulfate PS latex NPLs and sand grains, individual NPLs can be found attached to the sand surface due to mechanical retention within surface irregularities rather than adsorption. Both NPL types adsorb to clay minerals and metal oxide surfaces because of their heterogenous surface charges (E) [44].

In Geneva Lake water, amidine PS latex NPLs are stable and negatively charged due to the presence of DOM coating on their surfaces. Once confined in narrow pore space, amidine PS latex NPLs are forming heteroaggregates through cation bridges (G) or due to the DOM charge screening effects (I). The newly formed aggregates, still negatively charged, can attach to the sand surface through cation bridging (F) or be physically blocked in pores too narrow to enable their transport (G). Moreover, already-attached aggregates can provide more active adsorption sites for incoming particles (the ripening process) [45]. As for the sulfate PS latex NPLs, they are stable and negatively charged despite the cation and DOM coating and increased ζ potential. Once inside the sand filtration column, sulfate PS latex NPLs start to aggregate, and newly formed aggregates are physically strained inside pores (G). Individual sulfate PS latex NPLs can also attach to the negatively charged sand surface through cation bridging (F). Both NPL types can attach to clay minerals or metal oxide surfaces due to their heterogeneous surface charges (E) [44]. 

Ramirez et al. (2022) evaluated the removal efficiency of positively charged polystyrene nanoplastics during the sand and activated carbon filtration processes and reported that overall NPL removal was approximately 88% and retention was mostly controlled by adsorption and straining [24]. Moreover, they found that the coagulation process improves considerably the performance of sand filters, and in the presence of coagulant, the effective removal efficiency of nanoplastics increased from 54% to 99%. In our study, adsorption was limited, and aggregation and further aggregate straining contributed to NPL retention. However, the adsorption of NPLs to aged sand, i.e., sand coated with biofilm, can be more important. He et al. (2020) found that, in 10 mM NaCl solution, the breakthrough percentage for 0.02 µm NPLs, 0.2 µm MPLs, and 2 µm MPLs in clean quartz sand was approximately equal to 82%, 84%, and 88%, respectively, while their corresponding percentage in biofilm-coated sand decreased up to 58%, 17%, and 35%, respectively [46]. Moreover, it was reported that the growing biofilm on quartz sand surfaces reduced pore space, especially at the column inlet, changed the porosity of sand columns from 0.42 to 0.4, and contributed to the occurrence of a larger number of collisions between plastic particles and the filter bed. In addition, it was suggested that changes in quartz sand surface roughness induced by the presence of biofilm contributed more to plastic particle deposition than electrical interactions. Apart from that, He et al. (2020) evaluated the composition of EPS on the retention of plastic particles and found that, in both 10 mM and 50 mM NaCl solutions, the deposition of 0.2 µm MPLs was important and enhanced by three major EPS components, namely proteins, polysaccharides, and humic substances. The backwashing of the sand filters helps to improve their performance and eliminate adsorbed materials. The water produced during this procedure is then treated, organic materials are precipitated, and it is sent to urban wastewater networks for further treatment. 

## 4. Conclusions

In this work, the transport behavior of positively charged amidine PS latex NPLs and negatively charged sulfate PS latex NPLs in laboratory quartz sand columns saturated with two different types of water was studied.

We found that NPL retention inside quartz sand columns saturated with ultrapure water was more important for amidine PS latex NPLs than for sulfate PS latex NPLs. In the case of amidine PS latex NPLs, adsorption, caused by electrically attractive interactions between oppositely charged sand grains and NPLs, as well as aggregation and further straining of newly formed aggregates, are responsible for the removal of NPLs. For the sulfate PS latex NPLs, mechanical blockage and adsorption to the clay minerals and metal oxide surfaces caused NPL retention. Consequently, it is demonstrated that, in ultrapure water, surface charge plays a crucial role in NPL retention inside porous media. The obtained results are in accordance with other findings already presented in the literature, and they help to better understand changes in retention mechanisms and retention capacities when natural water fills pore space. 

The NPLs retention inside quartz sand columns saturated with Geneva Lake water was found to be important for both amidine PS latex NPLs and sulfate PS latex NPLs, even though the adsorption process plays a minor role in NPLs retention under environmental conditions. The role of the adsorption process is limited because of the repulsion forces present between negatively charged NPLs and negatively charged sand surfaces. Retention was mainly attributed to the presence of natural water components, which trigger NPL heteroaggregation. As a result, the newly formed heteroaggregates are further retained, to a great extent, in narrow pores. The obtained results demonstrate that the presence of metallic cations and DOM greatly improves the retention of both types of plastic when natural waters are considered. This is an important conclusion in the field of water treatment, especially for drinking water treatment plants based on conventional filtration units and using superficial freshwater resources to produce drinking water for the removal of nanoplastics, which suggests taking into account both the intrinsic filtration unit properties and raw water properties to evaluate the fate and removal efficiency of nanoplastics.

## Figures and Tables

**Figure 1 nanomaterials-14-00529-f001:**
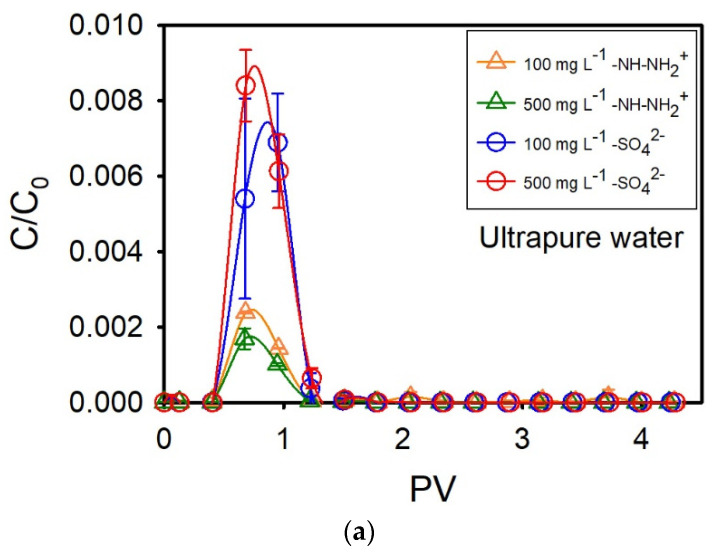
(**a**) Breakthrough curves of amidine PS latex NPLs (–NH–NH_2_^+^) and sulfate PS latex NPLs (–SO_4_^2−^) injected punctually at an initial concentration of 100 mg L^−1^ or 500 mg L^−1^ into quartz sand columns saturated with ultrapure water. One PV ranged between 69 mL and 73.0 mL. Z-average hyrodynamic diameter (**b**) and ζ potential (**c**) changes of amidine PS latex and sulfate PS latex NPLs present in the effluent water.

**Figure 2 nanomaterials-14-00529-f002:**
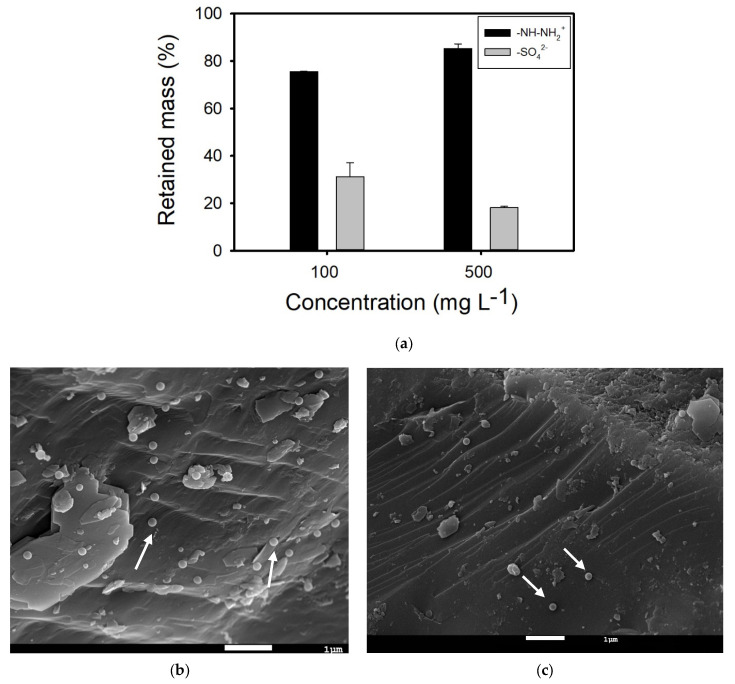
(**a**) Retained mass of different types of nanoplastics in quartz sand columns saturated with ultrapure water. (**b**) SEM image of amidine PS latex NPLs attached to the sand surface. Experimental conditions: [–NH–NH_2_^+^ NPLs] = 500 mg L^−1^, ultrapure water. (**c**) SEM image of sulfate PS latex NPLs attached to the sand surface. Experimental conditions: [–SO_4_^2−^ NPLs] = 500 mg L^−1^, ultrapure water. White arrows indicate NPLs.

**Figure 3 nanomaterials-14-00529-f003:**
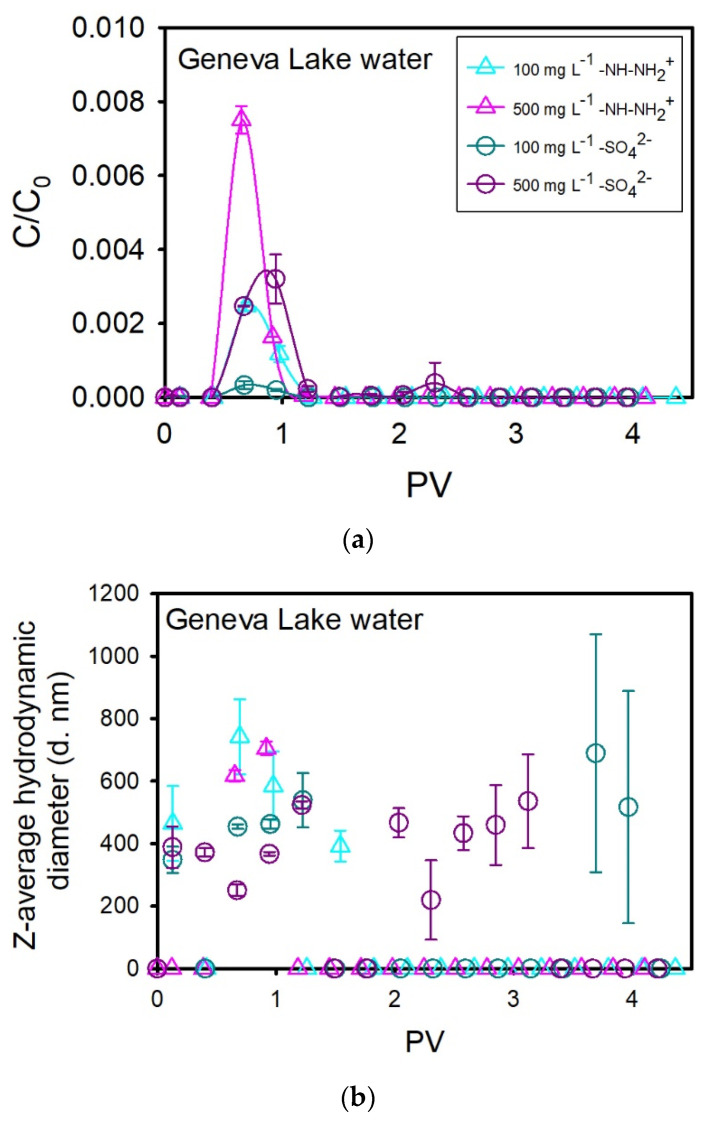
(**a**) Breakthrough curves of amidine PS latex NPLs (–NH–NH_2_^+^) and sulfate PS latex NPLs (–SO_4_^2−^) injected punctually at an initial concentration of 100 mg L^−1^ or 500 mg L^−1^ into quartz sand columns saturated with Geneva Lake water. One PV ranged between 69 mL and 73.0 mL. Z-average hyrodynamic diameter (**b**) and ζ potential (**c**) changes of amidine PS latex and sulfate PS latex NPLs present in the effluent water.

**Figure 4 nanomaterials-14-00529-f004:**
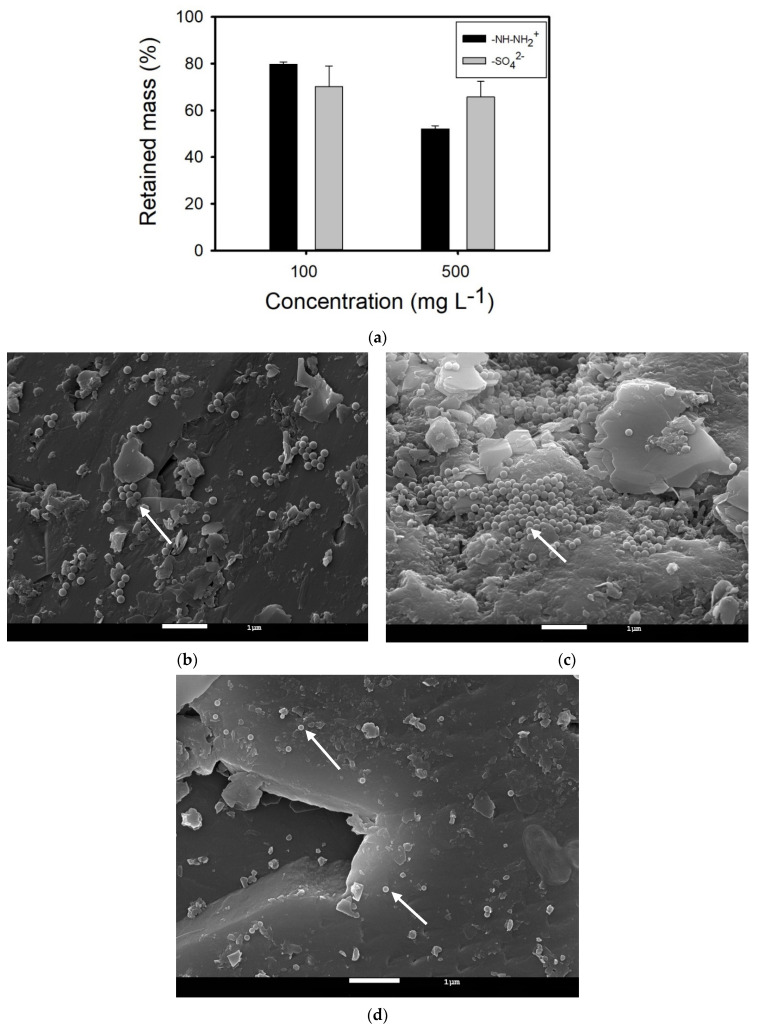
(**a**) Retained mass of different types of nanoplastics in quartz sand columns saturated with Geneva Lake water. (**b**,**c**) SEM image of amidine PS latex NPLs attached to the sand surface. Experimental conditions: [–NH–NH_2_^+^ NPLs] = 500 mg L^−1^, Geneva Lake water. (**d**) SEM image of sulfate PS latex NPLs attached to the sand surface. Experimental conditions: [–SO_4_^2−^ NPLs] = 500 mg L^−1^, Geneva Lake water. White arrows indicate NPLs.

**Figure 5 nanomaterials-14-00529-f005:**
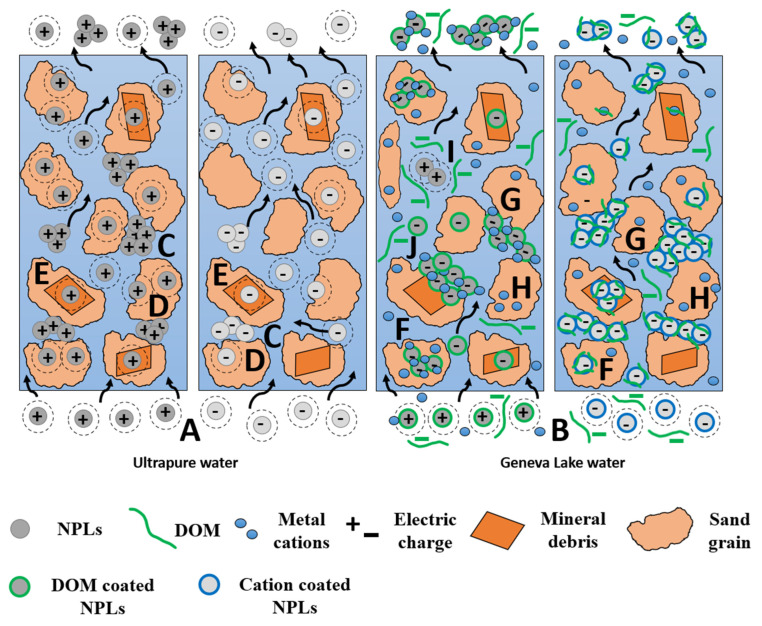
Summary of nanoplastics retention mechanisms in quartz sand filtration columns. A—stabilization due to the presence of repulsive forces; B—stabilization due to the presence of repulsive forces and despite the presence of DOM and cation coating; C—aggregation of NPLs confined in a limited pore space and further aggregate straining; D—adsorption to the sand surface; E—adsorption to the surface of clay minerals; F—adsorption to the sand surface via cation bridging, G—straining of formed aggregates; H—occupation of surface active sites by positively charged water components; I—aggregation due to DOM charge screening effects; J—ripening.

## Data Availability

Data will be made available on request.

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
