# Peer review of "Influence of Concentration, Surface Charge, and Natural Water Components on the Transport and Adsorption of Polystyrene Nanoplastics in Sand Columns"

_nanomaterials, 2024, doi:10.3390/nano14060529_

Round 1

Reviewer 1 Report

Comments and Suggestions for Authors

This is a very accurate and comprehensive work, and contributes to methods for addressing nanoplastics pollutants in water. I suggest publication and only have minor observations that could help improve the already high quality of this paper.

The term “nanoplastics” is never defined in the work. One might assume particles in the 10-100 nm, see also figure 4, but might be worth including a definition.

One of the main conclusions is that surface charge plays a crucial role in NPLs retention inside porous media. However, the authors should better stress why and how such conclusion represents a significant advancement from what is currently available in the literature on particles’ adsorption on porous media. This also applies to the use of quartz and sand as retentive matrices to capture pollutants. Some short additional comments/discussion in this sense would strengthen the article results.

Author Response

Comments and Suggestions for Authors

This is a very accurate and comprehensive work, and contributes to methods for addressing nanoplastics pollutants in water. I suggest publication and only have minor observations that could help improve the already high quality of this paper. The term “nanoplastics” is never defined in the work. One might assume particles in the 10-100 nm, see also figure 4, but might be worth including a definition.

Thank you very much for this comment. A whole paragraph including information about nanoplastics origin and nanoplastics definition was included in the new version of manuscript.

One of the main conclusions is that surface charge plays a crucial role in NPLs retention inside porous media. However, the authors should better stress why and how such conclusion represents a significant advancement from what is currently available in the literature on particles’ adsorption on porous media. This also applies to the use of quartz and sand as retentive matrices to capture pollutants. Some short additional comments/discussion in this sense would strengthen the article’s results.

Thank you very much for this comment. The conclusions were reformulated.

Reviewer 2 Report

Comments and Suggestions for Authors

The work entitled Influence of Concentration, Surface Charge and Natural Water Components on the Transport and Adsorption of Polystyrene Nanoplastics in Sand Columns is current topic, despite of the process results  are not promising. The work describes the influence of surface charges on the transport of nanoplastics (NPLs), in particular, positively charged polystyrene amidine latex (PS) and negatively charged sulfate PS latex NPLs

However, some questions/sugestions can be incorporated in the manuscript.

1. Key words must be different from words contained in the title.

2. “The styrene concentration in the provided suspension was equal to 50 g L-1 and a conversion into polystyrene was estimated to be 86 %.” Were these values previously estimated?

3. Have the pH of both ultrapure water and Lake Geneva water not been modified?

4. What reagent was used to adjust the pH? Does this influence the loads?

5. How to increase the adsorption capacity of the material? (The average adsorption capacity of sand grains for the removal of both NPLs types was very small and equal to 0.008 mg g-1 for NPLs concentration of 100 mg L-1)

5. If adsorption is limited, which process would be most suitable for removing NPLs?

6. regardless of whether the process is limited, authors must compare it with others.

Author Response

  1. Key words must be different from words contained in the title.

Thank you very much for this remark. Key words have been changed.

  1. “The styrene concentration in the provided suspension was equal to 50 g L-1 and a conversion into polystyrene was estimated to be 86 %.” Were these values previously estimated?

Thank you very much for this question. Amidine polystyrene latex nanoplastics solutions were synthesized and provided by an external laboratory. The given values were estimated by the provider and communicated to the authors of the present study. This information is now included in the manuscript.

  1. Have the pH of both ultrapure water and Lake Geneva water not been modified?

Thank you very much for this question. The pH of both ultrapure water and Geneva Lake water was measured before experiments. The pH of ultrapure water effluent and Geneva Lake water effluent was measured right after column experiments. No significant difference was observed between measured values. This information is now included in the manuscript.

  1. What reagent was used to adjust the pH? Does this influence the loads?

Only the pH of ultrapure water was adjusted using HCl or NaOH (Titrisol, Merck, Switzerland) during the preparation of dilutions. The pH of ultrapure water was equal to 5.7 ± 0.2, whereas the pH of Geneva Lake water was equal to 7.2 ± 0.2. As pH in both cases was stable during experiments, there were no additional adjustments. The influence of pH on nanoplastics transport was out of the scope of the present study because the particular interest was attributed to the influence of natural water components on nanoplastics transport and fate in porous media. These explanations were included in the improved manuscript. 

  1. How to increase the adsorption capacity of the material? (The average adsorption capacity of sand grains for the removal of both NPLs types was very small and equal to 0.008 mg g-1 for NPLs concentration of 100 mg L-1)

Even though the adsorption capacity of the sand used as a filtration medium in drinking water treatment plants is low, the overall retention capacity of the sand filters is very high. The processes of aggregation and straining play a more important role than adsorption. Moreover, other research showed that coagulation process taking place before sand filtration, and activated carbon filtration taking place right after sand filtration can largely improve nanomaterials removal [1]. In addition, some research showed that the presence of biofilm on sand surface can improve adsorption process [2]. The biofilm can develop on the sand surface naturally, after several weeks. The use of any chemical substances to improve sand adsorption capacity is not recommended as it is water destined for human consumption.

References:

  1. Ramirez Arenas, Lina, Stéphan Ramseier Gentile, Stéphane Zimmermann, and Serge Stoll, Fate and removal efficiency of polystyrene nanoplastics in a pilot drinking water treatment plant. Science of The Total Environment, 2022. 813: p. 152623.
  2. He, Lei, Haifeng Rong, Dan Wu, Meng Li, Chengyi Wang, and Meiping Tong, Influence of biofilm on the transport and deposition behaviors of nano- and micro-plastic particles in quartz sand. Water Research, 2020. 178: p. 115808.

  1. If adsorption is limited, which process would be most suitable for removing NPLs?

As said before, aggregation combined with straining are the most efficient in nanomaterials retention within porous media saturated with natural water.

  1. regardless of whether the process is limited, authors must compare it with others.

Thank you very much for this remark. The results were compared with other publications.

Reviewer 3 Report

Comments and Suggestions for Authors

The presented manuscript includes a study of the influence of concentration, surface charge and natural water components on the transport and adsorption of polystyrene nanoplastics in sand columns. 

The paper is of interest. The work looks interesting and clear, but some comments should be mentioned. 

1. Introduction. Please mention the toxicity of “Nanoplastic” to human health to prove the actuality of the work. Another point, “plastic” is not a scientific term, “polymer” is scientific. I can understand that it is widely used, so, I can suggest to mention this in the beginning, and then you can leave it in the text as “Nanoplastic”

2. Methodology. For granular filters in water treatment plants, a more common filtration parameter is filtration speed in “m/h”, could you provide this parameter as well? 

3. Methodology. It looks like any FTIR or IR analysis is missed to provide results of pristine and spent filtering material (sand) to additionally prove the presence of NPLs on the send surface.

4. Figs. 1, 3. Please add the unit to the X-axis

5. Figs. 2, 4. The SEM image of pristine sand is needed as well as EDS maps to show the difference and first prove the presence of NPLs on the surface. As well as FTIR or IR spectroscopy results.

6. Obtained results should be compared with published analogs.

7. Authors should describe the possibilities for regeneration of spent sand and the next treatment of washed waters.

Author Response

  1. Please mention the toxicity of “Nanoplastic” to human health to prove the actuality of the work. Another point, “plastic” is not a scientific term, “polymer” is scientific. I can understand that it is widely used, so, I can suggest to mention this in the beginning, and then you can leave it in the text as “Nanoplastic”

Thank you very much for this remark. The information about nanoplastics toxicity was included in the improved manuscript.

Regarding the use of term “nanoplastic” and “polymer”, we checked the definition, and according to “ISO/TR 21960:2020(en): Plastics — Environmental aspects — State of knowledge and methodologies”:

3.1

polymer

chemical compound or mixture of compounds consisting of repeating structural units created through polymerization

Note 1 to entry: In practice above 10 000 Dalton.

Note 2 to entry: Polymers comprise both plastics and elastomers. The latter is excluded from the scope of ISO/TC 61.

3.2

plastic

material which contains as an essential ingredient a high polymer (3.1) and which, at some stage in its processing into finished products, can be shaped by flow

Note 1 to entry: Plastics consists mainly polymers and minor contents of additives (3.7).

Note 2 to entry: Supplementary to the term “plastic”, “plastic product” is also used. According to ISO 472, a plastic product represents “any material or combination of materials, semi-finished or finished product that is within the scope of ISO/TC 61, Plastics”.

Note 3 to entry: Plastics comprise both thermoplastic (3.3) and thermoset (3.4) materials.

To conclude, “polymer” is a substance, which is present in plastics. Therefore, we prefer to use the term “plastic” or “nanoplastic”.

  1. For granular filters in water treatment plants, a more common filtration parameter is filtration speed in “m/h”, could you provide this parameter as well? 

Thank you for this remark. In drinking water treatment plant of Geneva, rapid filtration is used, and an average filtration speed is equal to 6 m/h. This information is now provided in the corrected manuscript.

  1. It looks like any FTIR or IR analysis is missed to provide results of pristine and spent filtering material (sand) to additionally prove the presence of NPLs on the send surface.

Thank you for this comment. FTIR or IR analysis were not used to prove the presence of NPLs on the sand surface, because of the very low concentration of adsorbed NPLs. However, SEM images were taken to show adsorbed NPLs. To better illustrate difference between pristine sand and sand sampled after filtration, additional SEM image of bare sand was included in Supporting Information.

  1. 1, 3. Please add the unit to the X-axis

Thank you for this remark. Pore volume (PV) is a kind of standardized method. One pore volume is defined as “the cross-sectional area of the column times the length times the porosity (ALn), i.e., the column’s capacity” (Contaminant Hydrogeology (3rd Edition) by Fetter, C. W. Boving, Thomas Kreamer, David). To avoid any confusion, the definition was included in the corrected manuscript, and the information about pore volume variation was included in the captions of the Figure 1 and 3.

  1. 2, 4. The SEM image of pristine sand is needed as well as EDS maps to show the difference and first prove the presence of NPLs on the surface. As well as FTIR or IR spectroscopy results.

Thank you for this remark. SEM image of pristine sand was added to Supporting Information. As discussed before EDS, FTIR or IR spectroscopy analysis was not performed because of the very low concentration of NPLs on the sand surface. 

  1. Obtained results should be compared with published analogs.

Thank you very much for this remark. The results were compared with other publications.

  1. Authors should describe the possibilities for regeneration of spent sand and the next treatment of washed waters.

The sand filters are not regenerated like activated carbon filters, but they backwashed. The water used for backwashing is then treated, organic matter is recovered and sent to urban wastewater networks. The procedure is now included in the corrected manuscript.

Round 2

Reviewer 3 Report

Comments and Suggestions for Authors

all comments were addressed